# TASK-DRIVEN DISCOVERY OF PERCEPTUAL SCHEMAS FOR GENERALIZATION IN REINFORCEMENT LEARNING

## ABSTRACT

Deep reinforcement learning (Deep RL) has recently seen significant progress in developing algorithms for generalization. However, most algorithms target a single type of generalization setting. In this work, we study generalization across three disparate task structures: (a) tasks composed of spatial and temporal compositions of regularly occurring object motions; (b) tasks composed of active perception of and navigation towards regularly occurring 3D objects; and (c) tasks composed of remembering goal-information over sequences of regularly occurring object-configurations. These diverse task structures all share an underlying idea of compositionality: task completion always involves combining reccurring segments of task-oriented perception and behavior. We hypothesize that an agent can generalize within a task structure if it can discover representations that capture these reccurring task-segments. For our tasks, this corresponds to representations for recognizing individual object motions, for navigation towards 3D objects, and for navigating through object-configurations. Taking inspiration from cognitive science, we term representations for reccurring segments of an agent's experience, "perceptual schemas". We propose *Feature Attending Recurrent Modules (FARM)*, which learns a state representation where perceptual schemas are distributed across multiple, relatively small recurrent modules. We compare FARM to recurrent architectures that leverage spatial attention, which reduces observation features to a weighted average over spatial positions. Our experiments indicate that our feature-attention mechanism better enables FARM to generalize across the diverse object-centric domains we study.

## 1 INTRODUCTION

Cognitive scientists theorize that humans generalize broadly with "schemas" they discover for regularly occurring structures within their experience (Minsky, 1979; Rumelhart, 1980; Rumelhart et al., 1986). Schemas are representations that capture common features over diverse aspects of the environment. For example, when we learn to drive, we learn schemas for common car types (such as sedans), common car motions (such as accelerating or stopping), and common car arrangements (such as a row of cars). Importantly, schemas are composable representations over portions of our observations. This allows us to recombine them in novel ways. For example, once we learn schemas for sedans, car motions, and rows of cars, we can recognize many rows of sedans moving in opposite directions—even if we've never seen this before.

While substantial progress has been made on developing deep reinforcement learning (deep RL) algorithms which can generalize, algorithms are typically limited to one type of generalization. For example, one algorithm will generalize to novel compositions of familiar shapes and colors (Higgins et al., 2017; Chaplot et al., 2018), whereas another algorithm will generalize to longer sequences of observed subtasks (Sohn et al., 2018; 2021; Brooks et al., 2021). In this work, we hypothesize that we can develop a *single* deep RL architecture that can exhibit *multiple* types of generalization if it can learn schema-like representations for regularly occurring structures within its experience. As a first step, we study learning schemas for perception (i.e. perceptual schemas) that support generalization within a diverse set of tasks and environments.

We study generalization across three diverse environments and task structures, each with their own regularly occurring structures (Figure 1). Across these environments, test tasks are novel compositions

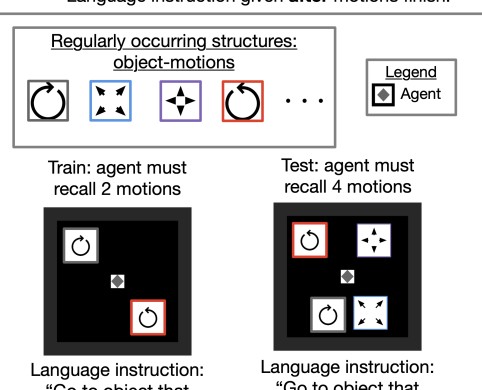

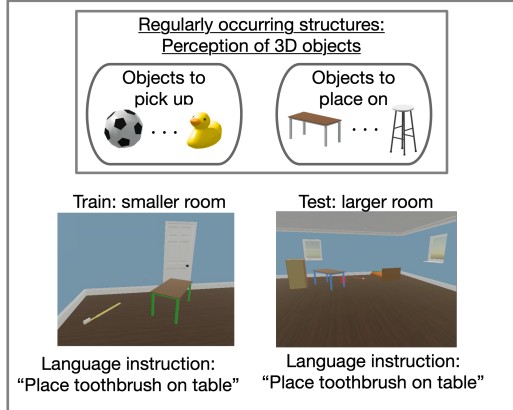

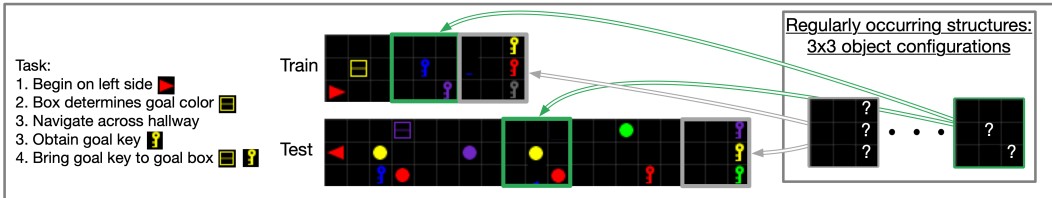

Figure 1: **We study three diverse compositional task structures in both 2D and 3D environments:** (a) composing memory of object-motions, (b) composing 3D objects with a larger environment, and (c) composing memory of goal information to longer tasks made of recurring object configurations. We test an agent's ability to generalize within a task structure by testing it on novel compositions of the regularly occurring structures the agent experienced during training. We hypothesize that an agent can successfully generalize at test time if it can discover perceptual schemas that capture these recurring structures. Videos of our agent performing these tasks: https://bit.ly/3kCkAqd.

of the regularly occurring structures the agent experiences during training. Generalization for the "Ballet task" involves recalling novel spatial and temporal compositions of regularly occurring object motions (Figure 1, a); generalization for the "Place X on Y task" involves generalizing active perception of and navigation towards regularly appearing 3D objects (Figure 1, b); and generalization for the "Keybox task" involves generalizing memory-retention of goal-information to larger environments composed of sequences of regularly occurring object-configurations (Figure 1, c). We hypothesize that discovering perceptual schemas for these regularly occurring structures will facilitate zero-shot generalization to tasks defined over novel compositions of these structures.

We propose *Feature Attending Recurrent Modules (FARM)*, a state representation learning architecture for discovering task-relevant perceptual schemas. FARM learns perceptual schemas that are *distributed* across multiple, smaller recurrent modules. To consider why this might be helpful, consider the benefits of using word embeddings. A word embedding can represent more information than a one-hot encoding of the same dimension because it can represent different patterns of word usage with the same dimension. Analogously, learning multiple modules enables FARM to represent different patterns of an agent's experience—i.e. different perceptual schemas—with the same module. To maximize the expressivity of the patterns a module can represent, each module employs a novel *dynamic feature attention* mechanism to dynamically attends to important features in the agent's observation. When combined with spatio-temporal features, our results suggest that the perceptual schemas FARM discovers capture diverse structures including object motions, 3D objects, and spatial-relationships between objects. To have the modules coordinate what they attend to, they share information using transformer-style attention (Vaswani et al., 2017).

Recent work indicates that spatial attention is a simple inductive bias for strong performance on object-centric vision tasks (Greff et al., 2020; Locatello et al., 2020; Goyal et al., 2020b;a). We compare FARM to recurrent architectures that employ spatial attention and contribute the following:
1. FARM, which combines dynamic feature attention with learning multiple recurrent modules (§3).

2. We show that FARM's components synergestically enable generalizing (a) recall to novel compositions of object motions; (b) active perception of 3D objects to larger environments; and (c) generalizing memory of goal-information to longer tasks filled with more distractors (§4).
3. We show that spatial attention—which reduces observation features to a weighted average over spatial positions—can be detrimental to reinforcement learning of our diverse object-centric tasks and interfere with the benefits that come from learning multiple recurrent modules.
4. Our analysis of the representations learned by FARM provide evidence that it learns perceptual schemas that are flexibly distributed across combinations of recurrent modules (§4.3.1).

## 2 RELATED WORK ON GENERALIZATION IN DEEP RL

While a large body of work has focused on studying generalization in deep RL (Kansky et al., 2017; Witty et al., 2018; Farebrother et al., 2018; Zhang et al., 2018b; Cobbe et al., 2019; Raileanu & Fergus, 2021), there has been less emphasis on studying generalization within diverse environments and task structures. One research direction has focused on generalizing to longer tasks, e.g. executing longer sequences (Oh et al., 2017; Zhang et al., 2018a; Lampinen et al., 2021) or executing novel subtask structures (Sohn et al., 2018; 2021; Brooks et al., 2021). Another direction has focused on generalizing to tasks with novel features, e.g., novel shape-color combinations (Higgins et al., 2017; Chaplot et al., 2018; Lee et al., 2020; Hill et al., 2020), novel backgrounds (Zhang et al., 2021; Agarwal et al., 2021), and novel distractors (Mott et al., 2019; Goyal et al., 2020b). We attempt to bridge these prior strands of research by developing a single architecture for (a) generalizing recall to novel compositions of object motions; (b) generalizing sequential perception of 3D objects to larger environments; and (c) generalizing memory of goal information to longer task lengths.

**Task-driven generalization.** Recent work which has shown that a diverse training curriculum can promote generalization (Tobin et al., 2017; Packer et al., 2018; Hill et al., 2020). This research inspired our task-driven approach to discovering generalizable "schema-like" representations. Additionally, our procedurally-generated KeyBox task follows previous research on using procedural level generalization for faster learning and generalization (Justesen et al., 2019; Jiang et al., 2021).

**Generalizing with feature attention**. Most similar to our feature-attention mechanism are the attention mechanisms by Perez et al. (2018); Chaplot et al. (2018). In particularly, Chaplot et al. (2018) showed that mapping language instructions to non-linear feature coefficients enabled generalizing to tasks specified over unseen feature combinations in a 3D environment. While FARM also learns non-linear feature coefficients, our work has two important differences. First, we develop a multi-head version where individual feature coefficients are produced by their own recurrent modules. This enables FARM to leverage this form of attention in settings where language instructions don't indicate what to attend to (this is true in $2/3$ of our tasks). Second, we are the first to show that feature attention facilitates generalizing recall of object dynamics (Figure 4.1) and generalizing memory-retention to larger environments (Figure 4.3).

**Generalizing with top-down spatial attention**. Most similar to FARM are the Attention Augmented Agent (AAA) (Mott et al., 2019) and Recurrent Independent Mechanisms (RIMs) (Goyal et al., 2020b). Both are recurrent architectures that produce top-down spatial attention over observations. Additionally, RIMs also uses a modular state representation produced by a set of recurrent modules. AAA showed generalization to unseen distractors and RIMs to more distractors than trained on. While follow-up work on RIMs has addressed problems such as learning type-specific update rules (Goyal et al., 2020a; Didolkar et al., 2021) and improved information sharing among modules (Mittal et al., 2020; Goyal et al., 2021), there hasn't been an emphasis on which components enable generalization in reinforcement learning across diverse domains. **The major difference between AAA, RIMs, and FARM is that FARM attends to an observation with feature attention as opposed to spatial attention.** Our experiments indicate that spatial attention can be an impediment to reinforcement learning of tasks defined by shape-color agnostic object-dynamics and 3D objects.

## 3 ARCHITECTURE: FEATURE ATTENDING RECURRENT MODULES

We study an agent that experiences a partial observation and task description $(x_t, \tau_t) \in \mathcal{X} \times \mathcal{T}$, takes an action $a_t \in \mathcal{A}$, and experiences resultant reward $r_t \in \mathbb{R}$. The agent needs a representation for state that enables learning a policy $\pi : \mathcal{S} \to \mathcal{A}$ that maximizes the expected reward from taking an action

at a state $\mathbb{E}_\pi[\sum_{t=0} \gamma^t R_{t+1}|S_t = a, A_t = a]$. Since the agent only observes observation-task-action-reward tuples $(x_t, \tau_t, a_t, r_t)$, it needs to learn a time-series representation mapping episode *histories* $(x_{\leq t}, \tau_{\leq t}, a_{<t}, r_{<t})$ to state representations $s_t$. The agent learns these representations with *Feature Attending Recurrent Modules (FARM)*.

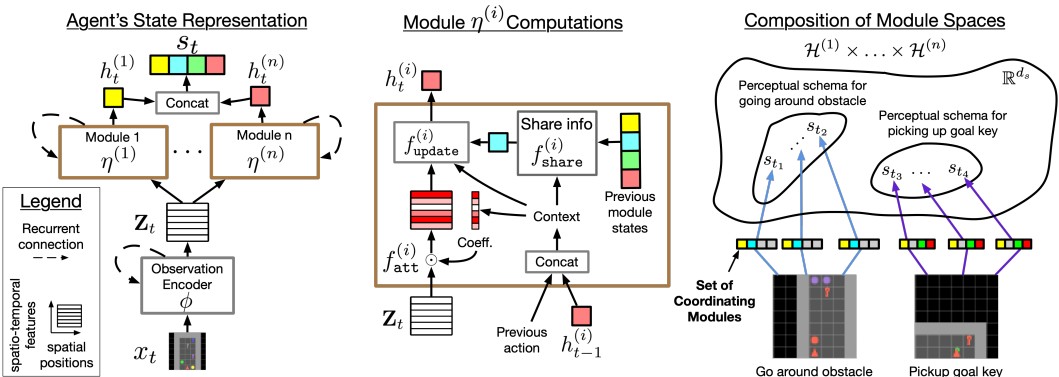

Figure 2: **FARM** learns representations for perceptual schemas that are distributed across $n$ recurrent modules $\{\eta^{(i)}\}$. To discover perceptual schemas, FARM exploits structured observation features $\boldsymbol{Z}_t = \phi(x_t, \boldsymbol{Z}_{t-1}) \in \mathbb{R}^{m \times d_z}$ that share $d_z$ spatio-temporal features across $m$ spatial positions. To capture diverse structures, each module uses feature attention to compute coefficients for important features: $f_{\text{att}}^{(i)}$. This allows a module to update with important features present across all spatial positions. The modules share information to coordinate what they represent with $f_{\text{share}}^{(i)}$.

**Exploiting structured observation features for generalization.** We assume that FARM can generalize if it represents task-relevant regularly occurring perceptual structures, i.e. perceptual schemas. To discovering perceptual schemas, we assume an observation encoder $\phi(\cdot)$ that produces observation features $\boldsymbol{Z}_t \in \mathcal{Z} \in \mathbb{R}^{m \times d_z}$ where $d_z$ features are shared across $m$ rows representing *different portions* of the observation. For an image, different rows might correspond to different spatial positions; for audio, to different frequency bands; or for robotic proprioception, to spatial information about different body parts. In this work, we focus on observations in the form of images.

**Feature Attending Recurrent Modules.** FARM learns representations for perceptual schemas that are distributed over $n$ recurrent modules $\{\eta^{(i)}\}$. Module $i$ uses a distinct initial module-state $h_0^{(i)} \in \mathcal{H}^{(i)}$ and the subsequently experienced observation-task-action-reward tuple $(x_t, \tau_t, a_t, r_t)$ to learn the following mapping, $\eta^{(i)} : \mathcal{Z} \times \mathcal{T} \times \mathcal{H}^{(i)} \times \mathcal{A} \times \mathcal{R} \to \mathcal{H}^{(i)}$. For convenience, we can define a **module context** $c_t^{(i)}$ as the concatenation of the previous module-state, the action chosen with it, the resultant reward, and the current task-description as $c_t^{(i)} = [\tau_t, h_{t-1}^{(i)}, a_{t-1}, r_{t-1}]$. The agent integrates this context $c_t^{(i)}$ with $\boldsymbol{Z}_t$ to compute $s_t$ using the collection of outputs from the modules:

$$s_t = [\eta^{(1)}(\boldsymbol{Z}_t, c_t^{(1)}), \dots, \eta^{(n)}(\boldsymbol{Z}_t, c_t^{(n)})] \tag{1}$$

$$= [h_t^{(1)}, \dots, h_t^{(n)}]. \tag{2}$$

**Module update rule.** Each module will use its context $c_t^{(i)}$ to (a) attend to some aspect of the observation and (b) to retrieve information from other modules. This manifests with two functions, (a) $f_{\text{att}}^{(i)}$ and (b) $f_{\text{share}}^{(i)}$, respectively. Defining $f_{\text{update}}^{(i)}$ as the module update-function, we model $\eta^{(i)}$ as the following composition of functions:

$$\eta^{(i)}(\boldsymbol{Z}_t, c_t^{(i)}) = f_{\text{update}}^{(i)}(c_t^{(i)}, \underbrace{f_{\text{att}}^{(i)}(\boldsymbol{Z}_t, c_t^{(i)})}_{\text{attend to observation}}, \underbrace{f_{\text{share}}^{(i)}(c_t^{(i)}, \{h_{t-1}^{(j)}\}_1^n)}_{\text{share information}}) \tag{3}$$

**Modules capture diverse perceptual structures with dynamic feature attention.** Our **first insight** was that modules can attend to high-level object-dynamics information if FARM learns a recurrent observation encoder $\boldsymbol{Z}_t = \phi(x_t, \boldsymbol{Z}_{t-1})$. By doing so, information about how feature-values shift between positions (i.e. feature dynamics) can be captured in the features of $\boldsymbol{Z}_t$. Our **second insight**

was that feature attention is an expressive attention mechanism for selecting observation information to update with. Each module predicts its own feature attention coefficients and applies them identically to all spatial positions in $\boldsymbol{Z}_t$ (Perez et al., 2018; Chaplot et al., 2018). We found it useful to linearly project the features before and after using shared parameters as in Andreas et al. (2016); Hu et al. (2018). The operations are summarized below:

$$f_{\text{att}}^{(i)}(\boldsymbol{Z}_t, c_t^{(i)}) = (\boldsymbol{Z}_t W_1 \odot \sigma(W_{\text{att}}^i c_t^{(i)}))W_2 \tag{4}$$

where $\odot$ denotes an element-wise product over the feature dimension and $\sigma$ is a sigmoid non-linearity. Equation 4 computes the degree to which important features are expressed across all positions. Since our features capture dynamics information, this allows a module to attend to dynamics (§ 4.1). When updating with equation 3, we flatten the output of equation 4 and give this as input to an RNN. Flattening leads all spatial positions to be treated uniquely and enables a module to represent aspects of the observation that span multiple positions, such as 3D objects (§ 4.2) and spatial arrangements of objects (§ 4.3). Since the feature-coefficients for the next time-step are produced with observation features from the current time-step, modules can *dynamically shift* their attention when task-relevant events occur (see Figure 6, c).

**Modules share information to coordinate what they represent.** Similar to RIMs Goyal et al. (2020b), before updating, each module retrieves information from other modules using transformer-style attention (Vaswani et al., 2017). We define the collection of previous module-states as $\boldsymbol{H}_{t-1} = \left[h_{t-1}^{(1)}; \ldots; h_{t-1}^{(n)}; \boldsymbol{0}\right] \in \mathbb{R}^{(n+1) \times d_h}$, where $\boldsymbol{0}$ is a null-vector used to retrieve no information. Each module retrieves information as follows:

$$f_{\text{share}}^{(i)}(c_t^{(i)}, \{h_{t-1}^{(j)}\}_1^n) = \text{softmax}\left(\frac{\left(c_t^{(i)}W_i^q\right)\left(\boldsymbol{H}_{t-1}W_i^k\right)^\top}{\sqrt{d_h}}\right)\boldsymbol{H}_{t-1}W_i^v \tag{5}$$

## 4 EXPERIMENTS

In this section, we study the following questions:

1. Can FARM enable generalization of memory-retention to novel spatio-temporal compositions of object-dynamics?
2. Can FARM generalize sequential active perception of 3D objects to larger environments?
3. Can FARM generalize memory-retention of goal-information to longer tasks composed of longer sequences of observed obstacles?

**Baselines.** Our first baseline is the canonical choice for learning state-representations, a **Long Short-term Memory (LSTM)** (Hochreiter & Schmidhuber, 1997). Our other two baselines — the **Attention Augmented Agent (AAA)** (Mott et al., 2019) and **Recurrent Independent Mechanisms (RIMs)** (Goyal et al., 2020b) — also employ top-down attention over observation features. However, they both employ transformer-style attention (Locatello et al., 2020; Vaswani et al., 2017) to dynamically attend to *spatial positions* in the

| Method | Observation Attention | Modular State |
|---|---|---|
| LSTM | ✗ | ✗ |
| AAA | Spatial | ✗ |
| RIMs | Spatial | ✓ |
| FARM (Ours) | Feature | ✓ |

Table 1: Comparison of baselines.

observation; whereas, we dynamically attend to *features shared across all spatial positions*. RIMs, like FARM, composes state with a set of recurrent modules.

**Implementation details.** We implement our recurrent observation encoder, $\phi$, as a ResNet (He et al., 2016) followed by a Convolutional LSTM (ConvLSTM) (Shi et al., 2015). We implement the update function of each module with an LSTM. We used multihead-attention (Vaswani et al., 2017) for $f_{\text{share}}^{(i)}$. We trained the architecture with the IMPALA algorithm (Espeholt et al., 2018) and an Adam optimizer (Kingma & Ba, 2015). We tune hyperparameters for all architectures with the "Place X next to Y" task from the BabyAI environment (Chevalier-Boisvert et al., 2019) (§ C.2). We keep most hyperparameters fixed across our tasks. Our main change is to make the RNNs employed by each architecture larger for the KeyBox task. For details on hyperparameters, see §B.

### 4.1 GENERALIZING MEMORY-RETENTION TO NOVEL COMPOSITIONS OF OBJECT-DYNAMICS

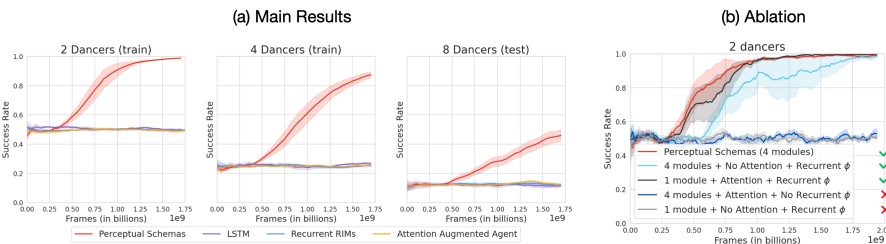

Figure 3: **FARM enables generalizing recall to novel spatio-temporal compositions of object motions**. We present the success rate means and standard errors computed using 5 seeds. (a) Only FARM is able to go above chance performance for each setting. (b) Given recurrent features, we find that learning perceptual schemas with *either* multiple modules *or* feature attention enables learning recall of spatio-temporal compositions. These results suggest that spatial attention removes the benefits of learning multiple modules when learning to recall object-dynamics.

We study this with the "Ballet" grid-world (Lampinen et al., 2021) shown in Figure 1 (a). The agent is a white square in the middle of the grid. There are $m$ other "ballet-dancer" objects that move with a distinct object-motion for 16 time-steps. There are 15 object-motions. The dances appear in sequence for 16 time-steps with a 48-time-step delay in between. Training always consists of seeing $m = \{2, 4\}$ dancers; testing always consists of seeing $m = \{8\}$ dancers. After all dancers finish, the agent is given a language instruction and it must go to the correct ballet-dancer. The agent gets a reward of 1 if it goes to the correct dancer, and 0 otherwise. All shapes and colors are randomized making the dynamics the only feature that indicates the goal-object. We study the success rate of an agent. A poorly performing agent will obtain chance performance, $1/m$. All agents learn with a sample budget of 2 billion frames and use approximately 7M parameters.

We present the training and generalization success rates in Figure 3. We learned recurrent observation features with RIMs and AAA for a fair comparison. We find that only FARM is able to obtain above chance performance for training and testing. In order to understand the source of our performance, we ablate using a recurrent observation encoder, using multiple modules, and using feature-attention. We confirm that recurrent encoder is required. Interestingly, we find that either using multiple modules or using our feature-attention enables task-learning, with our feature-attention mechanism being slightly more stable. We hypothesize that using a recurrent encoder with our attention allows for storing information about how object-motions are shifting position in a way that mitigates interference.

(Lampinen et al., 2021) showed that a hierarchical transformer architecture was able to learn and generalize recall of sequential object-motions in the Ballet task. However, their architecture uses 6 distinct 8-head hierarchical attention operations and 8 MLP layers, making it much deeper and more computationally intensive than ours. Their architecture had 13.3M parameters whereas ours has 7.1M. We show that when using recurrent observation features, simply learning top-down feature attention or using multiple RNNs can enable learning this task.

### 4.2 GENERALIZING SEQUENTIAL ACTIVE PERCEPTION OF 3D OBJECTS TO LARGER ENVIRONMENTS

We study this with the 3D Unity environment from Hill et al. (2020) shown in Figure 1 (b). This environment tests an agent's ability to generalize sequential composition of active perception of 3D objects to larger environments. The agent is an embodied avatar in a room filled with task objects and distractor objects. The agent has 46 discrete actions and has a navigation step-size of $.15m$. The agent receives a language instruction of the form "*X on Y*" —e.g., "toothbrush on bed" and receives a reward of 1 if it completes the task and 0 otherwise. We partition objects into two sets as follows: pickup-able objects $O_1 = A \cup B$ and objects to place them on $O_2 = C \cup D$. During training the agent sees $A \times D$ and $B \times C$ in a $4m \times 4m$ room with 4 distractors, along with $A \times C$ and $B \times D$ in a $3m \times 3m$ room with 0 distractors. We test the agent on $A \times C$ and $B \times D$ in a $4m \times 4m$ room with 4 distractors. The training curriculum also includes "Go to X" and "Lift X". For details see § E.3.

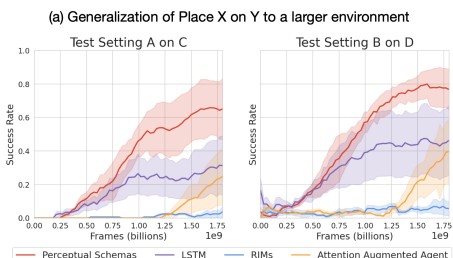 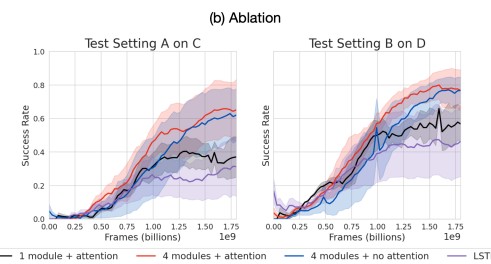

Figure 4: **FARM enables generalizing sequential active perception of 3D objects to a larger environment.** We present the success rate means and standard errors computed using 3 seeds. (a) FARM generalizes best. (b) Our performance benefits come mainly from learning multiple modules, though feature attention slightly improves performance and lowers variance. These results suggest that spatial attention interferes with reinforcement learning of 3D objects.

We present the generalization success rate in Figure 4. All architectures and ablations had approximately 5M parameters. We find that baselines which used spatial attention learn more slowly than an LSTM or FARM. Additionally both models using spatial attention have poor performance until the end of training where AAA begins to improve. FARM achieves relatively good performance, achieving a success rate of $60\%$ and $80\%$ on the two test settings, respectively. We hypothesize that our feature attention mechanism can more easily attend to 3D objects spanning multiple positions as opposed to spatial attention which uses a softmax that may saturate on individual positions.

### 4.3 GENERALIZING MEMORY-RETENTION TO LONGER TASKS

In order to study this, we create the multi-level "keybox" environment depicted in Figure 1 (c). This environment tests an agent's ability to learn to remember goal-information for increasingly long time-horizons. The agent maximizes reward by completing as many levels as it can in an episode. Each level in this environment is a hallway with a single box and a **key of the same color**. The hallway for level $n$ consists of a length-$n$ sequence of $w \times w$ environment subsections. Each subsection contains $d$ distractor objects that can either be a ball of any color or keys of non-goal colors. The agent and the box always starts in the left-most subsection and the goal key always starts in the right-most subsection. Each time the agent succeeds, it is teleported to the next level and gets a reward of $n/n_{\max}$ where $n_{\max}$ is the maximum level the agent can complete. We set $n_{\max} = 10$ during training. The agent has $50n$ time-steps to complete a level. The distractors pose an additional challenge of obstructing the agent's path. This further challenges memory since they have to be incorporated into the state-representation to predict actions but should not overwrite the goal information. **We study two generalization settings**: a **densely populated setting** that has subsections of width $w = 3$ with $d = 2$ distractors and **sparsely populated setting** that has subsections of width $w = 5$ with $d = 4$ distractors. In the dense setting, once the agent fails, it restarts in level $n \in [1, n_{\text{done}}]$, where $n_{\text{done}}$ is the highest level the agent has completed. In the sparse setting, we found we needed to always restart the agent on level 1 to enable learning. All architectures used approximately 7.5M parameters.

We present the maximum training level reached and generalization success rates in Figure 5. We find that all methods achieve comparable training results. In the dense setting, we see an LSTM quickly overfits in both settings. All architectures with attention continue to improve in generalization performance as they continue training. In the dense setting, we find that FARM tends to generalize better (by about $20\%$ for AAA and about $30\%$ for RIMs). In the sparse setting, both RIMs and an LSTM fail to generalize above $30\%$. FARM better generalizes than the AAA for level 20 but gets comparable performance for level 30. In some ways, this is our most surprising result since it is not obvious that uniformly attending to features across all spatial positions should help with this task. In the next section we study possible sources of our generalization performance.

#### 4.3.1 ANALYSIS OF REPRESENTATIONS FOR REGULARLY OCCURRING EVENTS

We study the representations FARM learns for categories of regularly occurring perceptual events. We collect 2000 generalization episodes in level 20. We segment these episodes into 6 categories of regularly occurring perceptual events: pickup ball, drop ball, pickup wrong key, drop wrong

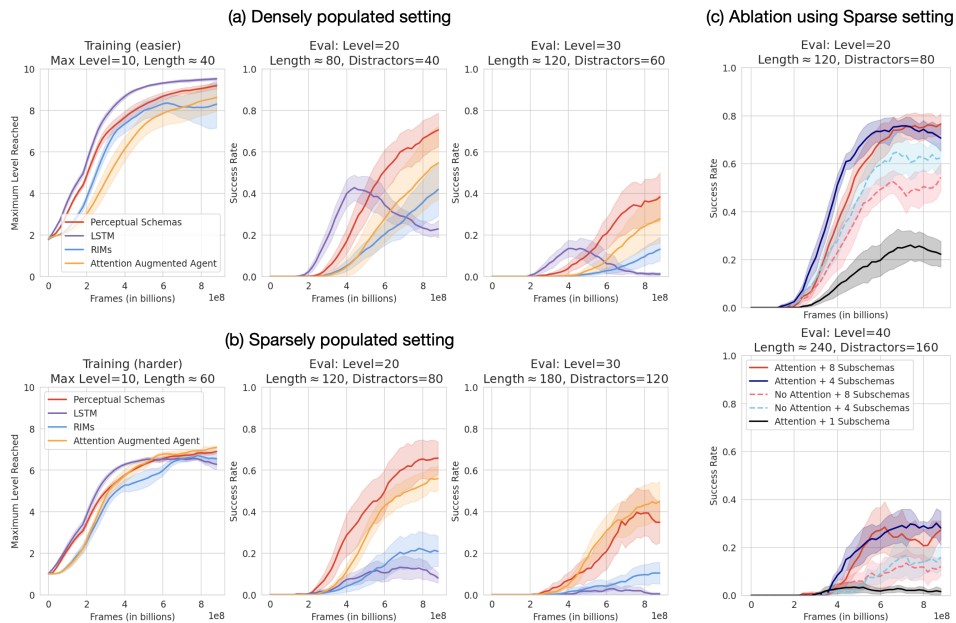

Figure 5: **FARM enable generalizing memory-retention of goal-information to longer horizons of obstacles than trained on**. On the left-most panel of (a,b), we present the maximum training level reached by each agent. All others panels in (a-c) show the generalization success rate. For all quantities, we present the mean and standard error computed using 10 seeds. (a) All architectures achieve comparable training performance. FARM better generalizes to longer hallways with more distractors in the densely populated setting. (b) FARM and AAA get similar generalization performance in the sparsely populated setting. (c) Using multiple modules and feature attention both improve generalization. These results suggest that spatial attention interferes with generalization benefits of learning multiple modules while feature attention acts synergistically.

key, pickup correct key, and drop correct key. In order to understand how FARM represents these categories, we study time-series of the L2 norm of each module LSTM-state and of their attention coefficients. For reference, we also show the L2 norm for the entire episode.

We present a subset of results in Figure 6. For all results, see §D. It is worth noting that programmatically extracting all task-relevant events for analysis is challenging. For example, we were able to observe salient state activity from module 0 when the agent moved around obstacles but found no way to extract this signal. We show an example in Figure 6 a. While some modules are selective for different recurring events such as attending to goal information (Figure 6, a-c), it seems that the perceptual schema used for recurring events are generally distributed across multiple modules. We hypothesize that this enables FARM to leverage some modules to store goal-information and other modules to represent and navigate around obstacles in increasingly long environments. This is further supported by our ablation where we find that having 4 (8.2M params) or 8 (7.5M params) modules significantly outperforms using a single large module (7.5M params) (Figure 5, (c)). Feature attention consistently improves performance.

## 5 DISCUSSION AND CONCLUSION

We have presented, FARM, a novel recurrent state representation learning architecture. Our results show that a single architecture can support the following three disparate types of generalization: (a) generalizing recall to novel spatio-temporal compositions of object-motions; (b) generalizing active perception of 3D objects to larger environments; and (c) generalizing memory of goal-information to longer sequences of obstacles. Our results suggest that learning feature attention and learning multiple recurrent modules can act synergistically, with different components being more helpful in different settings. For example, feature attention enables more stable learning of object-motions (Figure 3)

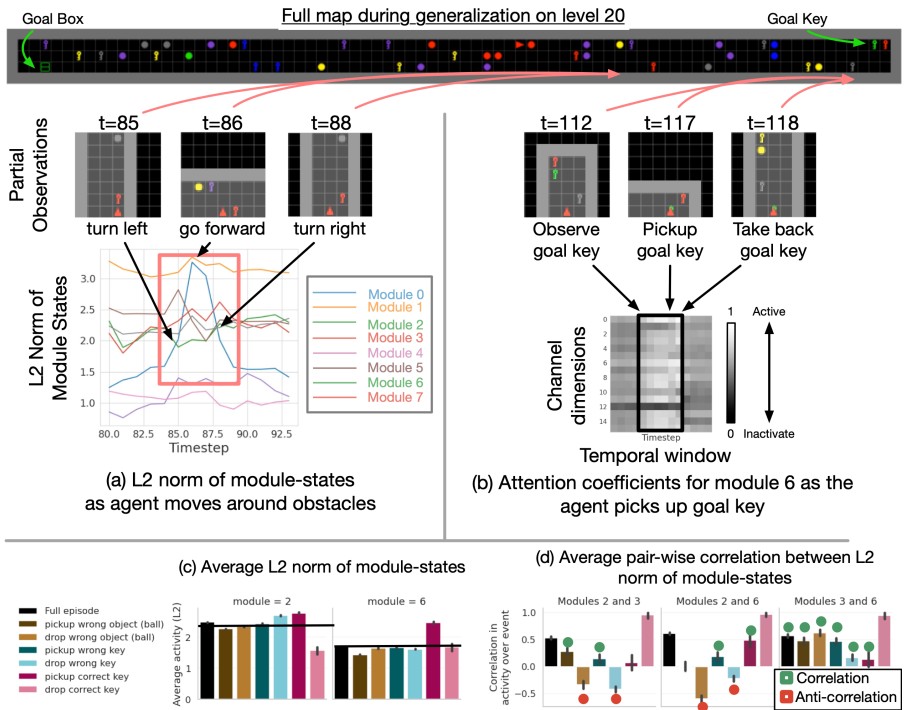

Figure 6: **While (a-c) show that individual modules have salient activity for different events, (d) shows that different combinations of modules coordinate their activity for different events. This indicates that event representations are distributed flexibly across the modules.** (a) Module 0 commonly exhibits salient activity when the agent moves around an obstacle. (b) Module 6 shows selective activity for representing goal information. (c) Module 6 activates its attention coefficients as the agent picks up the goal key. (d) Some modules activate together (i.e. correlate) to represent some recurring events but activate in opposite directions (i.e. anti-correlate) to represent other recurring events. For example, modules 2 and 6 correlate for picking up the correct key but anti-correlate for dropping the wrong object. This is similar to when neurons in word embeddings correlate for some words (e.g. man and king), but anti-correlate for other words (e.g. man and woman). These results suggest that FARM is distributing representations for perceptual schemas across the modules in complicated ways rather than representing individual schema with separate modules. Videos of the state-activity and attention coefficients: https://bit.ly/3qCxatr.

while multiple modules better enables generalization with 3D objects (Figure 4). Importantly, neither component hinders the other and an architecture with both succeeds on a broader set of tasks. Together, the two enable the discovery of representations for shape-color agnostic motion (§4.1), for active perception of 3D objects (§4.2), and for spatial relationships between objects (§4.3, §C.3).

**Feature attention vs. spatial attention.** While our dynamic feature attention mechanism is flexible, FARM learns an update function that treats all positions in the attention output as unique (equation 4). A major advantage of spatial attention is that the resultant representation is *spatially-invariant*, which may sometimes be more useful for a policy. For example, we find that RIMs has better generalization to an unseen number of distractors (§C.2). Interestingly, using spatial attention seems to interfere with the generalization benefits of learning multiple recurrent modules. An interesting future direction might be to explore combining both forms of attention for a more flexible policy.

**RNNs vs. Transformers.** Another interesting future direction involves discovering perceptual schemas with transformers (Vaswani et al., 2017). When used as memory for an RL agent, Transformers have shown strong performance for representing state over longer sequences but show weaker performance for reactive behavior (Parisotto et al., 2020). An interesting next-step might be to use a hierarchical architecture where FARM provides representations for relatively short sequences. These could be used as primitive observation-representations for a transformer-like architecture. Such an architecture might enable discovering perceptual schemas over very long time-horizons.

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
