# OpenReview forum: "Task-driven Discovery of Perceptual Schemas for Generalization in Reinforcement Learning"
_ICLR.cc/2022/Conference — ICLR 2022 Submitted_

### Official Review · Reviewer_4R1W · 2021-10-30

**Correctness:** 3
**Technical Novelty And Significance:** 2
**Empirical Novelty And Significance:** 3
**Recommendation:** 5
**Confidence:** 4

**Main Review:**

Strengths:

Interesting set of diagnostic tasks/environments have been developed to study generalization of deepRL agents along various axes such as object types, object numbers, sequence length of task solution etc.

Experiments showing qualitative analysis of the subschema representations learned by the CPS model are interesting. More such qualitative studies/ablations or visualizations (various attention coefficients during an episode, subschema activity during an episode? etc.), could further help dissect and gain insight into the various components of the proposed CPS model and their roles in the generalization performance.


Weaknesses:

Proposed model and architectural novelty is quite similar in nature to several of the follow-up works on RIMs [1, 2, 3, 4]. Discussion and comparison with highly relevant follow-up work on RIMs [1, 2, 3, 4] which also propose solutions to the same problems of i) how to share information among modules [1, 3] ii) learning type-specific update rules [2, 4] has not been addressed or evaluated. How does the proposed novel feature attention mechanism and the general CPS model architecture benefit over other solutions proposed by these works [1, 2, 3, 4] to the similar problems?

Another issue is with the lack of relevant baseline models being used to judge the benefits of the proposed CPS model architecture and its novel feature attention mechanism. The RIMs baseline model struggles to learn on several tasks (Figure 3 (a) (b), Figure 4), sometimes shows even worse performance than the vanilla LSTM. RMC [5] which models relational information among memory slots and uses this information to update the memory slots using key-value attention would make for a good and (relatively) simple baseline model to compare against.


Results:

It’s rather interesting that the vanilla LSTM is able to generalize better than RIMs on a few tasks such as Dancers (Fig. 3.a) and 3D Place X on Y task (Fig. 4). This is surprising since RIMs have inductive biases that explicitly favour modularity and compositionality in representation over the vanilla LSTM, which are beneficial to generalize on these tasks. Could the authors offer
some explanation for this behavior?

Are the total number of learnable parameters always kept the same across CPS model variants with 1/2/4/8 subschemas in all the experiments?

The paper would improve if the authors could add/move some of the visualizations from the Appendix to the section 4.3.1 to expand this discussion and qualitative analysis on the extent of specialization and modularity in representations/behavior captured by various subschemas. This would further strengthen the author’s hypothesis of “learning n subschemas, each with 1/n parameters of a single monolithic module….. encourages the discovery of regularly occurring structures.”

I didn’t quite understand how to interpret the visualizations on the pairwise correlation between subschemas in Fig. 6.b. While they show that subschemas interact with one another during an event, shouldn’t this interaction be rather sparse? If the subschemas are to specialize to unique recurring input patterns in an episode, shouldn’t just one subschema ideally activate only for the recurring events (ex: pickup key, drop key etc.) it has specialized to represent? If so, then how can there be a high correlation between subschema activations over an event? Does that indicate that multiple subschemas are modelling the same recurring pattern/event amongst them? Could the authors provide some clarification on this?

Writing/Presentation:

How are the attention coefficients (alpha^i) in the f_att function computed?

$f_{att}^{(i)} (x_t, e_{t-1}^{(i))} = (Z_t … )$. What is $e_{t-1}^{(i)}$? Do you mean $c_{t-1}^{(i)}$?

“We used multihead-attention for $f_{message}$”. What is $f_{message}$? It was not defined earlier.

Please include highly relevant information about certain key agent/training/environment hyperparameters in the main text that is needed to judge the validity of empirical results/ablations. The reading experience can greatly improve if one doesn’t have to constantly jump from main text to Appendix to gather information needed to judge the experimental settings and validity of results.

“... note that “Place A on C” and “Place B on D” are both in our training curriculum …” wording is confusing. Does this mean that the agent has seen the same task structure in training but not with the same objects to pick and place?

Minor typos:

“Reoccuring” -> “recurring”?

“Pickupable” -> “pickable”

“Policy-state” -> “state”



[1] Mittal et. al, “Learning to Combine Top-Down and Bottom-Up Signals in Recurrent Neural Networks with Attention over Modules”, ICML 2020.

[2] Goyal et. al, “Factorizing Declarative and Procedural Knowledge in Structured, Dynamical Environments”, ICLR 2021.

[3] Goyal et. al, “Coordination Among Neural Modules Through a Shared Global Workspace”,

[4] Goyal et. al, “Neural Production Systems”, https://arxiv.org/abs/2103.01937

[5] Santoro, Adam, et al. "Relational recurrent neural networks.", NeurIPS 2018.


**Summary Of The Paper:**

The paper proposes a modular state representation learning architecture called Composable Perceptual Schemas to discover regularly recurring patterns in its observations into “schemas” and process them in a modular and factorized manner. The proposed model is tested for generalization in the deepRL setting along various axes such as varying object types, object numbers, distractors, task solution lengths etc. and shows promising improvements over other recurrent baseline models.


**Summary Of The Review:**

The paper develops an interesting suite of diagnostic tasks/environments to study generalization in deepRL agents. While the technical contributions and results are interesting, it is difficult to fully judge the merits of the proposed model and the main contribution of the feature attention mechanism if it is not referenced, discussed or evaluated in the context of related prior works [1, 2, 3, 4] that study the similar problems and propose similar solution methods. Further, the paper would greatly improve with more qualitative analysis into the CPS model’s components and their role in its generalization performance. For these reasons I’m giving a score of 5. I’m willing to increase the score if my concerns are addressed.

---

> ### Author Response · Authors · 2021-11-17
> **Response**
>
> Thank you for your useful comments and suggestions. Among other things, we have used them to improve our analysis and better describe how the recurrent modules interact to represent the agent’s experience. We respond to individual concerns below.
>
>
>
> **Relation to RIM follow-ups**
>
> Thank you for pointing out this work. We have amended our related works to include it
>
> > While follow-up work on RIMs has addressed problems such as learning type-specific update rules (Goyal et al., 2020a; Didolkar et al., 2021) and improved information sharing among modules (Mittal et al., 2020; Goyal et al., 2021), there hasn’t been an emphasis on which components enable generalization in reinforcement learning across diverse domains.
>
> We hope that this work contributes to this growing body of literature on how to leverage structured architectures for learning and generalization in RL.
>
>
>
> **Suggestion to use RMC as a baseline**.
>
> Thank you for this suggestion. Given the time-limit of this rebuttal, we do not have time to implement, test and search over hyperparameters for this baseline. We hope that our ablations of feature-attention are informative to the utility of learning multiple recurrent modules without any attention.
>
>
>
> **Why does an LSTM outperform RIMs?**
>
> While learning multiple modules is a strong benefit of RIMs, our ablations indicate that adding spatial attention removes these benefits. We believe that spatial attention is prohibitive to reinforcement learning in the domains we study. Please see updated Figures 3, 4, and 5.
>
>
>
> **Number of parameters across model variants**
>
> Across all experiments and ablations, parameters were kept approximately equal. We have updated the text to include the number of parameters for each task. Below is an example for the keybox task.
>
> > This is further supported by our ablation where we find that having 4 (8.2M params) or 8 (7.5M params) modules significantly outperforms using a single large module (7.5M params) (Figure 5, (c)).
>
>
>
> **Adding more ablations and visualizations**.
>
> We have added an ablation to Section 4.2 and expanded the analysis in section 4.3.1. Please refer to figure 6 to see activity of modules along with activity coefficients. Here are videos of the module activity and attention coefficients during the keybox task: [https://bit.ly/3qCxatr](https://bit.ly/3qCxatr)
>
>
>
> **Shouldn't modules have sparse interaction?**
>
> This is a good question. We were not clear in the text previously. In complex domains, we expect multiple modules to collaborate to represent the agent's experience. Thus, we aim for perceptual schemas to be distributed across n modules instead of being specialized representations produced by one module. We have updated the introduction, Figure 2, the text in section 3, and the analysis discussion in Figure 6 to reflect this.
>
>
>
> **How is attention computed?**
>
> We have updated eq 4 to be more clear about this:
>
>
> $f^{(i)}\_{\tt att}(Z\_t, c^{(i)}_{t}) = (Z\_t W\_{1} \odot \sigma(W^{i}\_{\tt att} c^{(i)}\_{t}) ) W\_{2}$
>
>
> **Including important agent/training/environment hyperparameters in main text**
>
> We have added text on the number of parameters each architecture had. We are happy to include more information to make the main text easier to read. Could you please share some examples to give us a better idea of what we can move to improve clarity?
>
>
>
> **3D place tasks**
>
> We were not clear with our description here. We have changed the text as follows:
>
> > During training the agent sees A×D and B×C in a 4m×4m room with 4 distractors, along with A×C and B×D in a 3m×3m room with 0 distractors. We test the agent on A×C and B×D in a 4m × 4m room with 4 distractors.
>
> The agent had seen those subtasks but not in that room size with that number of distractors.

---

> > ### Comment · Reviewer_4R1W · 2021-11-29
> > **Response to authors**
> >
> > I'd like to thank the authors for taking my feedback into account in the revised manuscript.

---

### Official Review · Reviewer_jU4K · 2021-11-01

**Correctness:** 3
**Technical Novelty And Significance:** 2
**Empirical Novelty And Significance:** 3
**Recommendation:** 6
**Confidence:** 4

**Main Review:**

The focus of this paper is on addressing three particular types of generalization in RL as listed above. A commonality between them is that test tasks are novel compositions of regularly occurring "structures" the agent experiences during training. In the first case, these are the object-motions, and in the second and third case the visual objects. The difference in generalization behavior in the second and third cases is less clear to me since both can be viewed as requiring generalization to a larger environment and novel object compositions.
Regardless, it is interesting to consider a representation learning architecture that benefits each of these settings.

The main hypothesis, thus, is that a modular representation is desirable since individual modules can learn to capture these separate structures. This is achieved in CPS by having multiple different RNNs process the input separately and exchange information with one another to form a state representation, similar to how this is done in RIMs. Unlike in RIMs, however, here the input attention is over the features produced by an observation encoder, and attention is to all positions, while RIMs attend to particular spatial positions.
The motivation for this is not quite clear to me, other than that it is now possible to capture information about an object that covers multiple spatial positions. A potential downside is that it is no longer possible to attend to a particular location.

The technical contribution of CPS and technical novelty compared to prior work is therefore highly limited, especially since similar forms to the multiplicative input attention across positions explored here have also been explored previously, eg. in FiLM (as is also correctly pointed out in the related work section). Regarding the motivation of CPS based on "perceptual schemas", the connection is not quite clear to me either. In essence, the modularity in CPS and the reason why this may be beneficial for generalization is identical to how this is motivated in prior work such as RIMs. Therefore, it is not clear why modules or RIMs are now rebranded as perceptual subschema's, and one should view the learned representation as a perceptual schema. What exactly is new here that warrants a connection to "schemas" and a rebranding of existing terminology from previous papers?

The empirical contribution of CPS does appear substantial since in all the considered domains CPS is shown to outperform the baselines, which include RIMs. However, there are several experimental results regarding CPS and baselines that I would like to see clarified:

* On the Ballet task it is found that either feature attention or modules are sufficient for generalization. At the same time, RIMs and AAA are shown to perform very poorly. The only difference (minus minor architectural differences) of CPS with no attention to RIMs is the lack of a recurrent encoder in case of RIMs. Is the hypothesis thus that the lack of recurrent features is mainly responsible for these baselines performing poorly? If this is the case then it seems that this task is quite adversarial to allow for a fair comparison to RIMs. Would it be possible to endow RIMs with similar capabilities (eg. a recurrent input encoder) as a more representative baseline? Similarly, although AAA does have access to the recurrent input features it also performs poorly compared to the LSTM. What is the hypothesis for this behavior? Can it be tested?

* Similarly on the 3D Unity environment, the supposedly stronger baselines of AAA and RIMs struggle greatly compared to CPS and the LSTM. In this case it is not clear that recurrent input features are needed, and so the previous comparison to CPS with 1 slot + attention and to CPS with multiple slots and no attention would be particularly revealing here. I would like to see this comparison here or some other experiment to better explain the performance gap to CPS and AAA.

The multi-level keybox experiments are more revealing since it is now demonstrated how there is a clear benefit to having attention and modules, which was not the case for Ballet and couldn't be observed on 3D Unity. At the same time, it can now also be observed how AAA performs quite well and the performance gap to this baseline is small. In general, although there is often a substantial gap to the baselines it is not quite clear to me yet which components are contributing the most, and therefore how significant the empirical contribution in reality is. The analysis of representations for regularly occurring events is interesting and indeed suggests some specialization among the modules as was also previously found in RIMs.

Finally, regarding the overall clarity and presentation of the work, I find that it could be improved substantially. In particular, the choice of different types of generalization seems rather arbitrary and it is unclear why these particular types are significant. Regarding the second and third generalization regime it is also not quite clear what the dimensions are that are being composed over. To me it seems that they should be the objects, yet the text talks about composable segments as well. The correspondence of perceptual schemas to parts of the learned representation is also a bit confusing. At first, it is suggested that perceptual schemas take on things like car motions, sedans, etc. yet these are the structures that are captured by the modules, which are referred to as subschemas. My question thus is, which are the schemas, the individual modules or their composition? Regarding the method details, I would strongly encourage the authors to include some pseudo-code for this algorithm, and better highlight what are the novel contributions. It also isn't clear how the policy network is structured to leverage the underlying modularity of the representation, which seems like an important choice to discuss. The text below eq. 4 about the potential capabilities of using attention in this way in combination with eq 3. was not clear to me, and it would be good to provide concrete examples of how the things outlined there could be achieved.

Some minor comments:

* There is a mention of f_message, which is mentioned nowhere else. I assume this is an artifact of a previous version.

* I encountered several typo's, which a spell checker should reveal for you as well.


**Summary Of The Paper:**

This paper proposes Composable Perceptual Schemas (CPS), a modular state representation learning architecture for reinforcement learning that combines modular RNNs as in RIMs (Goyal et al., 2020) with a dynamic feature attention mechanism. It is hypothesized that this lets CPS exhibit multiple different kinds of generalization encountered previously in the literature: (1) generalization to environments given by novel compositions of known object motions, (2) generalizing active perception of 3D objects to larger environments, and (3) generalizing goal-directed behavior and memory retention to larger environments.

The idea of CPS is as follows:

* A recurrent observation encoder produces observation features for different parses of the input observation (here spatial locations in the image) to yield a feature matrix. Here each row corresponds to a different parse.

* A number of recurrent modules (here called "subschemas"), each process the observation features + context (action, reward, task encoding) and the previous state of all other modules using dynamic attention. From this, an updated state is computed using an LSTM.

* The dynamic attention for observations + context proceeds by scaling the observation features at all positions with a vector of learned attention coefficient. In this way, feature selection can take place (but shared across positions).

* The dynamic attention for other modules is similar to the implementation in RIMs using transformer-style attention.

CPS is evaluated on three environments and compared to several baselines. In general, it can be seen to exhibit the desired generalization behavior and outperform the baselines. Finally, some evidence is presented that subschemas are selective for semantically meaningful aspects of the input (eg. goal information, distractor information, etc.).

**Summary Of The Review:**

This paper presents a method for learning modular representations for RL that allow for generalization in three regimes. The technical novelty is highly limited, but the empirical contribution is promising. At the same time, I would like to see a number of the presented results clarified and some additional analysis done to understand why the baselines perform so poorly and which parts of the proposed method contribute the most in different regimes.

---

> ### Author Response · Authors · 2021-11-17
> **Response part 1**
>
> Thank you for your insightful comments and questions. We have used it to make the presentation of our tasks and method clearer, and to better present how different components contribute to different learning regimes. We respond to individual concerns below.
>
>
>
> **What is the difference in generalization in the 3D place and keybox tasks?**
>
> In KeyBox task, the agent has to generalize memory of goal information to navigate through 2x and 3x the number of distractors as during training. In the 3D place task, goal information is given at every time-step in the form of synthetic language so there is no memory challenge here. Here, the agent only has to generalize sequential navigation between 3D objects to a larger environment with the same number of distractors as during training. We have ammended the caption to figure 1 as follows (b=3D place, c=keybox)
>
> > (b) composing 3D objects with a larger environment, and (c) composing memory of goal information to longer tasks made of recurring object configurations.
>
> We have also changed the figure to more explictly explain the keybox task, and added videos of the agents performing the task: [https://bit.ly/3kCkAqd](https://bit.ly/3kCkAqd)
>
>
>
> **Why do we want to attend to features instead of positions?**
>
> What you said is exactly correct, "it is now possible to capture information about an object that covers multiple spatial positions". More generally, it's possible to capture information over _any_ aspect of the observation that covers multiple positions. One example of this is capturing information about a 3D objects the agent observes. This can span many spatial positions. While the module cannot attend to spatial positions, the resultant policy is still a function of all spatial positions, so information that is specific to a location can still be used for decision-making. We updated the methods below eq 4 as follows:
>
> > When updating with equation 3, we flatten the output of equation 4 and give this as input to an RNN. Flattening leads all spatial positions to be treated uniquely and enables a module to represent aspects of the observation that span multiple positions, such as 3D objects (§ 4.2) and spatial arrangements of objects (§ 4.3).
>
>
>
> **Why rebrand modules as subschemas and why name the representations "perceptual schemas"?**
>
> We changed the architecture to "Feature-Attending Recurrent Modules" and updated Figure 2 to better relate to existing work. We agree that we share some of the motivation and benefits of RIMs in terms of modularity. RIMs relies on spatial attention to select its inputs. Unfortunately, we found that spatial attention seems to hinder the benefits of recurrent modules during reinforcement learning. We changed the introduction to this effect:
>
> > Recent work indicates that spatial attention is a simple inductive bias for competitive performance on object-centric vision tasks (Greff et al., 2020; Locatello et al., 2020; Goyal et al., 2020b;a; Zoran et al., 2020).
> >
> > 3.  We show that spatial attention—which reduces observation features to a weighted average over spatial positions—can be detrimental to reinforcement learning of our diverse object-centric tasks and interfere with the benefits that come from learning multiple recurrent modules.
>
> We introduced perceptual schemas for two reasons. (a) We were seeking an architecture that could enable generalization in reinforcement learning over domains with diverse visual challenges. (b) We were targetting test tasks that involved combining regularly occurring visual structures. Cognitive scientists posit that humans can combine representations in diverse domains by learning "schemas".
>
>
>
> **Why does our architecture outperform RIMs and AAA on the Ballet task?**
>
> You rightly pointed out that one difference between our architecture without attention and RIMs is the lack of a recurrent encoder. There is another important difference: RIMs and AAA both use spatial attention. We hypothesize that spatial attention is the prohibiting factor for learning this task. To isolate this, we have added a recurrent observation encoder to RIMs (which we call "Recurrent RIMs"). The results are updated in Figures 3 and 7. We found that providing RIMs with recurrent features did not improve performance on this task. This further supports the notion that spatial attention is making reinforcement learning of these tasks more difficult. We note that we moved the parallel ballet results to the appendix to expand our analysis and include visualizations of the agent during generalization.

---

> > ### Author Response · Authors · 2021-11-17
> > **Response part 2**
> >
> >
> > **Ablation on 3D unity environment**
> >
> > We have included this in Figure 4. We have also added the following text:
> >
> > > Our performance benefits come mainly from learning multiple modules, though feature attention slightly improves performance and lowers variance. These results suggest that spatial attention interferes with reinforcement learning of 3D objects.
> >
> >
> >
> > **How does policy network use modularity of the representation?**
> >
> > To keep things simple, we simply concatenate the representations of each module. We have updated Figure 2 to reflect this.
> >
> >
> >
> > **Improving clarity of text below eq 4**.
> >
> > We have clarified the text as follows:
> >
> > > Since our features capture dynamics information, this allows a module to attend to dynamics. When updating with equation 3, we flatten the output of equation 4 and give this as input to an RNN. Flattening leads all spatial positions to be treated uniquely and enables a module to represent aspects of the observation that span multiple positions, such as 3D objects (§4.2) and spatial arrangements of objects (§4.3). Since the feature-coefficients for the next time-step are produced with observation features from the current time-step, modules can dynamically shift their attention when task-relevant events occur (see Figure 6, c).
> >
> >
> >
> > **Pseudo-code for algorithm.**
> >
> > We have put pseudo-code in Appendix section A. We hope this and our improvements to section 3 improve the clarity of the method and showcase its simplicity.

---

> > > ### Comment · Reviewer_jU4K · 2021-11-30
> > > **Reply**
> > >
> > > I thank the authors for their detailed response and the changes made to the paper. I find that the paper has improved substantially along several axes, which is why I will raise my score by 1 point to a 6.
> > >
> > > In general, I still find the paper borderline due to the inconsistent empirical results and the difficulty of extracting clear generalizable insights beyond the setting/method considered here (which itself is only motivated somewhat satisfactory). In that sense, I strongly agree with the sentiment echoed by the other reviewers, including the remaining issues raised by reviewer Y5Kp in their reply.

---

### Official Review · Reviewer_Y5Kp · 2021-11-02

**Correctness:** 2
**Technical Novelty And Significance:** 2
**Empirical Novelty And Significance:** 2
**Recommendation:** 5
**Confidence:** 4

**Main Review:**

It is nice that this paper is motivated by representing schemas from cognitive science, and to my knowledge this paper provides a novel architecture attempting to do so. From what I understand, the architecture aims for each module to specialize to represent a particular spatial configuration of a scene, and different modules might represent different spatial configurations.

I have the following concerns:

1. It was difficult for me to take away from this paper a set of general principles that can be applied to future research. I would expect that the paper either (1) goes the scientific route: takes the best theory of schemas from cognitive science and show how their architecture is entailed by that theory, in which case the experiments would help us refute alternative theories of schemas proposed in cognitive science and give us insight about the nature of schemas from the abstract perspective, or (2) goes the engineering route: proposes an architecture that gives much better performance on the evaluation tasks, in which case the experiments would help us understand what engineering principles are useful to incorporate into future methods. This paper does not do (1), so I would expect it to do (2). However, the experimental results suggest that CPS is often similar to the LSTM baseline (section 4.1, 4.2), and gets the same performance as the AAA baseline (section 4.3). Furthermore, for the Ballet task, the paper does not compare against the method (HCAM) in the paper that proposed the Ballet task (https://arxiv.org/pdf/2105.14039.pdf), so it is unclear whether CPS offers insight on the best way to solve such a task.
2. It was not sure what to conclude based on the analysis in section 4.3.1 and Appendix C. For example, could it be possible that the plots of Figure 6b, Figure 8, and Figure 9 could have just as well have been generated by just looking at the random initial weights of the schemas? Given that the authors are motivated by modeling schemas that model independent aspects of the task, what measurement could the authors make to evaluate whether the schemas have specialized to learning something meaningful about different aspects of the task, in a way that would be distinguishable from the measurement take on randomly initialized schema modules?
3. I am unsure what to conclude from Figure 3c. It seems that having the ablations of having 4 subschema modules without attention and having only 1 subschema module with attention performs similarly to the full CPS. If this is the case, how would I evaluate the CPS as a contribution, if ablations that remove the core parts of the method (>1 schema, or feature attention) still perform similarly well? If the authors hypothesize that "We hypothesize that learning n subschemas, each with 1/n parameters of a single monolithic module, reduces overfitting and encourages the discovery of regularly occurring structures," does Figure 3c actually refute their hypothesis?
4. In various occasions the paper makes claims about the abilities of the CPS module that are not sufficiently tested. For example, the authors state in 4.1 that " Learning this task tests a recurrent architecture’s ability to recognize and maintain separate,
independent dynamics in an agent’s state representation" --> can the authors please conduct the experiment that tests whether the schema modules actually learn independent dynamics with an appropriate metric of independence? As another example, in section 3: " Since the feature-coefficients for the next time-step are produced with observation features from the current time-step, subschemas can dynamically shift their attention when task-relevant events occur" --> can the authors please conduct the experiment that evaluates whether this shift in attention is actually true? If there is a way to visualize how each schema module's attention to different features changes over time, and match that attention activation to semantically meaningful events in the episode, that would give much needed insight on how the method works. Another example is, in section 3: "To encourage specialization, each subschema has $\approx \frac{1}{n}$ the parameters a single module would have." --> can the authors please conduct an experiment that measures the degree of specialization with an appropriate evaluation metric, for various different numbers of parameters of the subschema? It is unclear to me that simply restricting the parameter count is enough to induce specialization.

**Summary Of The Paper:**

This paper proposes the Composable Perceptual Schemas recurrent architecture. This recurrent architecture is decomposed into $n$ LSTMs, each of which the authors call a _schema module_. At each time-step:
1.  a recurrent encoder produces a set of $m$ feature vectors of dimension $d_z$, one vector for a different spatial location
2.  the task information $\tau_t$, the previous reward $r_{t-1}$, and previous action $a_{t-1}$ gets concatenate with the previous hidden state $h_{t-1}^{(i)}$ of module $i$. The authors call this concatenation the _context_ $c_t^{(i)}$.
3. The update rule of each schema module produces the new hidden state $h_{t}^{(i)}$ from three inputs:
    1. The output of $f_{att}$: A single soft mask of length $d_z$ is computed from $c_t^{(i)}$ and is applied to each of the $m$ feature vectors.
    2. $c_t^{(i)}$
    3. The output of a multi-head attention layer, where the queries are the $c_t^{(i)}$, and the keys and values are the previous hidden states of the schemas, along with the null vector: $[h_{t-1}^{(1)}, ... h_{t-1}^{(n)}, 0]$

The main difference between CPS and the baseline recurrent architectures (AAA and RIMs) is in step 1 of the update, how $f_{att}$ is computed: instead of attending over the different spatial locations CPS attends over the same dimensions of all the spatial features.

The authors apply their architecture to three generalization scenarios:
1. " Can CPS enable generalization of memory-retention to novel spatial and temporal (spatiotemporal) compositions of object-dynamics?" --> tested on the Ballet task from https://arxiv.org/pdf/2105.14039.pdf. CPS performs slightly better than the LSTM when generalizing to different spatial compositions, and much better than all baselines when generalizing to different spatial and temporal compositions.
2. " Can CPS generalize sequential active perception of 3D objects to larger environments?" --> tested on the "Place X on Y" task from https://arxiv.org/pdf/1910.00571.pdf. CPS learns faster than the baselines.
3. "Can CPS generalize goal-oriented behavior and memory-retention to environments composed of longer sequences of observed object-configurations?" --> tested on a custom KeyBox environment written in the BabyAI framework. CPS performs similarly to the AAA baseline.






**Summary Of The Review:**

Overall, while this paper is tackling an important question, its execution could be substantially improved. It was not clear what the contribution of the paper is from a scientific or engineering perspective. The analysis could be more informative about what the modules are actually learning. The central claim of the necessity of feature attention and multiple schema modules seems to be refuted from Figure 3c. The paper needs to either temper its claims or provide empirical evaluation for its claims. More details about these concerns are in the Main Review. It could be that I may have misunderstood something, in which case I would happily engage in discussion with the authors.

== AFTER REBUTTAL ==
I thank the authors for their response and additional experiments. I have raised my score to a 5, and I especially appreciated the clarifications and the qualitative analysis. After reading the rebuttal and the revised paper, one concern I still have is that it is difficult to glean a single coherent empirical message from the paper: on some tasks FARM performs the baselines (Figure 3, 4), on other tasks FARM performs similarly (e.g. Figure 5b, 7, 8). Perhaps this could be something that could be fixed through the writing, but overall the case for the following questions could be made stronger: (1) Why these three types of generalization rather than some other types of generalization? What is the underlying property that all these three types of generalization have in common, and why is this property important for AI research? (2) What is the explanation for why the proposed method must be the method for tackling tasks with this underlying property?

Regarding (1), it would strengthen the paper's overall message to identify a single unifying characteristic of the problems considered, because at the moment the paper is written in a way that highlights the disparate nature of these generalization tasks, but without an underlying theme connecting these generalization tasks, they seem to be chosen in an ad hoc way. Regarding (2), this could be addressed with stronger empirical results.

---

> ### Author Response · Authors · 2021-11-17
> **Response part 1**
>
> Thank you for your constructive comments. We have used them to make the engineering insights more apparent in the introduction and discussion, to clarify our claims throughout the text, and to better present our analysis. We respond to individual concerns below.
>
>
>
> **Results suggest CPS is similar to the LSTM baseline (section 4.1, 4.2), and gets the same performance as the AAA baseline (section 4.3)**
>
> We have added the following text to Figure 3 to clarify performance on the sequential setting on the Ballet task.
>
> > Only FARM is able to go above chance performance for each setting.
>
> Additionally, in 4.2 an LSTM gets 40% worse on (A on C) and 30% worse on (B on D). (We note that we moved the parallel ballet results to the appendix to make more room for our analysis).
>
>
>
> **Our architecture vs. HCAM. What insight is offered?**
>
> Thank you for pointing this out. FARM has a lower computational complexity than HCAM. We have confirmed with the authors of the paper that their architecture is more challenging to implement and more compute intensive than our own. We have added the following text
>
> > (Lampinen et al., 2021) showed that a hierarchical transformer architecture was able to learn and generalize recall of sequential object-motions in the Ballet task. However, their architecture uses 6 distinct 8-head hierarchical attention operations and 8 MLP layers, making it much deeper and more computationally intensive than ours. Their architecture had 13.3M parameters whereas ours has 5.1M. We show that when using recurrent observation features, simply learning top-down feature attention or using multiple RNNs can enable learning this task.
>
> Another benefit of our architecture is its ability to perform well in diverse domains. While it may not perform as well as methods designed for those domains, our experiments suggest that our architecture can achieve good reinforcement learning performance and generalization of (a) recall of object-motions, (b) active perception of 3D objects, and (c) memory-retention of goal information. We have added the following to our discussion:
>
> > Our results show that a single architecture can support the following three disparate types of generalization: (a) generalizing recall to novel spatio-temporal compositions of object-motions; (b) generalizing active perception of 3D objects to larger environments; and (c) generalizing memory of goal-information to longer sequences of obstacles.
>
>
>
> **Might Results from figures {6b, 8, 9} be due to random initial weights? What should we conclude?**
>
> Regarding figure 6b, we have updated the text to better explain how to analyze the text. We have re-run this analysis using randomly initialized weights and added this to the appendix (Figure 10 and Figure 11). The reader should take away from these figures that pairs of modules are learning to coordinated in a task-specific manner. We have ammended the text for Figure 10 (comparing average activity of trained and random weights) as follows:
>
> > We find that when weights are trained on the task, some modules are selective for different events. For example, Module 4 is selective for “drop wrong key” and module 6 is selective for “pickup correct key”. When we use random weights, we see that all modules have the same activity for all events. This indicates that they have not learned any task-specific activity.
>
> We have ammended the text for Figure 11 (comparing correlation of trained and random weights) as follows:
>
> > When looking at trained weights (left), we find that pairs of modules will have high correlation on some events and high anti-correlation on other events. For example, modules 7 and 2 correlate for drop wrong object and drop wrong key but anti-correlate pickup wrong object and pickup correct key. If we look at random weights (right), we see that pairs of modules will either fully correlate (modules 6 and 2), fully anti-correlate (modules 6 and 1), or have weak/no correlation (modules 6 and 4) for events. Importantly, we don’t see a significant mixture correlation and anti-correlation like we see with trained weights. This suggests that the random weights have less task-specific learning/uses by the agent.

---

> > ### Author Response · Authors · 2021-11-17
> > **Response part 2**
> >
> >
> >
> > **Does Figure 3c refute hypothesis that dividing parameters reduces overfitting and encourages discovery of regularly occurring structures?**
> >
> > We do not believe Figures 3c refutes our hypothesis. Figures 3c and 5c both suggest that feature attention and > 1 module both enable discovery as we hypothesized. While dividing parameters does encourage discovery in the Ballet task, it seems that feature attention by itself is enough to achieve this. On the other hand, in the KeyBox task, both are needed for our performance. It seems that both components have larger benefits in different settings. An architecture that incorporates both design choices is capable of succeeding in a broader set of tasks. To make this clear, we have revised the discussion of the paper as follows
> >
> > > Our results suggest that learning feature attention and learning multiple recurrent modules can act synergistically, with different components being more helpful in different settings. For example, feature attention enables more stable learning of object-motions (Figure 3) while multiple modules better enables generalization with 3D objects (Figure 4). Importantly, neither component hinders the other and an architecture with both succeeds on a broader set of tasks.
> >
> > Note that feature attention learns to recall 2 object-motions with very small variance in roughly 1 billion frames of experience whereas learning multiple modules has higher cariant until about 2 billion frames (i.e. double the time).
> >
> >
> >
> > **Misunderstanding about learning independent dynamics during ballet task**
> >
> > Thank you for pointing this out. We did not mean to intend that individual modules were learning independent dynamics. We have removed this text from the description in the first paragraph of 4.1.
> >
> >
> >
> > **Experiment that evaluates whether a shift in attention occurs**
> >
> > We added these results to figure 6b. We show the attention coefficients of module 6 as the agent picks up the goal key in the keybox task. We find that the coefficients increase as the agent approaches the goal key and immediately decrease once the agent has the key and is returning to the goal box. You can find raw videos here: [https://bit.ly/3qCxatr](https://bit.ly/3qCxatr).

---

> > > ### Comment · Reviewer_Y5Kp · 2021-11-30
> > > **Response to Rebuttal**
> > >
> > > I thank the authors for their response and additional experiments. I have raised my score to a 5, and I especially appreciated the clarifications and the qualitative analysis. After reading the rebuttal and the revised paper, one concern I still have is that it is difficult to glean a single coherent empirical message from the paper: on some tasks FARM performs the baselines (Figure 3, 4), on other tasks FARM performs similarly (e.g. Figure 5b, 7, 8). Perhaps this could be something that could be fixed through the writing, but overall the case for the following questions could be made stronger: (1) Why these three types of generalization rather than some other types of generalization? What is the underlying property that all these three types of generalization have in common, and why is this property important for AI research? (2) What is the explanation for why the proposed method must be the method for tackling tasks with this underlying property?
> > >
> > > Regarding (1), it would strengthen the paper's overall message to identify a single unifying characteristic of the problems considered, because at the moment the paper is written in a way that highlights the disparate nature of these generalization tasks, but without an underlying theme connecting these generalization tasks, they seem to be chosen in an ad hoc way. Regarding (2), this could be addressed with stronger empirical results.

---

### Author Response · Authors · 2021-11-17
**Shared response to all reviwers**

We thank all reviewers for taking the time to read the paper and provide insightful comments. We have tried to incorporate them into our paper as best we can. Most notably, we now state our contributions more clearly in the introduction. We improved the clarity of figure 2, expanded on the takeaways from all figures, and expanded our analysis to show module activity, attention coefficients, and better explain the correlation analysis. Please read below for responses to general points made by all reviewers.


**What engineering insights can future work learn from this paper?**

We have clarified this in the text. We have 3 insights:

1.  Feature attention aids generalization in RL

2.  In contrast, spatial attention seems to hurt RL

3.  Learning multiple recurrent modules also aids generalization in RL

We have added the following text at the end of the introduction to make this clear.

> Recent work indicates that spatial attention is a simple inductive bias for strong performance on object-centric vision tasks (Greff et al., 2020; Locatello et al., 2020; Goyal et al., 2020b;a). We compare FARM to recurrent architectures that employ spatial attention and contribute the following:
>
> 1. FARM, which combines dynamic feature attention with learning multiple recurrent modules (§3).
>
> 2. We show that FARM’s components synergistically enable generalizing (a) recall to novel compositions of object motions; (b) active perception of 3D objects to larger environments; and (c) generalizing memory of goal-information to longer tasks filled with more distractors (§4).
>
> 3. We show that spatial attention—which reduces observation features to a weighted average over spatial positions—can be detrimental to reinforcement learning of our diverse object-centric tasks and interfere with the benefits that come from learning multiple recurrent modules.
>
> 4. Our analysis of the representations learned by FARM provide evidence that it learns perceptual schemas that are flexibly distributed across combinations of recurrent modules (§4.3.1).



**Providing more visualizations and analysis in the main paper**

1.  We have created videos of the agents performing each task: [https://bit.ly/3kCkAqd](https://bit.ly/3kCkAqd)

2.  We updated figure 6 to include module activity and attention coefficients during different events in the KeyBox task.

3.  We have created videos showing the module activity and attention coefficients during the keybox task: [https://bit.ly/3qCxatr](https://bit.ly/3qCxatr)



**Improvements on clarity regarding what perceptual schemas are and to what degree modules in our architecture specialize:**

- We have removed claims about subschema specializing (e.g. sentence below eq 2). Neural networks are notoriously difficult to interpret and we believe that we must do more work before making such a strong claim.

- In order to not unnecessarily rebrand prior terminology, we have changed the name "subschema" to "recurrent module".

- We have changed the architecture name from "composable perceptual schemas" to "Feature Attending Recurrent Modules (FARM)" to better communicate the computational basis of our architecture.

- Finally, we have reframed the text to emphasize that the architecture learns representations for perceptual schemas that are distributed across multiple recurrent modules as opposed to being produced by 1 module. We have revised figure 2 and added visuals showing how perceptual schemas can be distributed across the recurrent modules. We have also added the following text

  > FARM learns perceptual schemas that are distributed across multiple, smaller recurrent modules. To consider why this might be helpful, consider the benefits of using word embeddings. A word embedding can represent more information than a one-hot encoding of the same dimension because it can represent different patterns of word usage with the same dimension. Analogously, learning multiple modules enables FARM to represent different patterns of an agent’s experience—i.e. different perceptual schemas—with the same module.

---

> ### Author Response · Authors · 2021-11-17
> **Shared response (part 2)**
>
> **Why might modules be active at once?**
>
> In complex domains, we expect using multiple modules enables the agent to represent different patterns in its experience with the same module. This is similar to how one dimension of a word embedding can represent different patterns in word usage. We have updated Figure 6 to address this question.
>
> > Some modules activate together (i.e. correlate) to represent some recurring events but activate in opposite directions (i.e. anti-correlate) to represent other recurring events. For example, modules 2 and 6 correlate for picking up the correct key but anti-correlate for dropping the wrong object. This is similar to when neurons in word embeddings correlate for some words (e.g. man and king), but anti-correlate for other words (e.g. man and woman). These results suggest that FARM is distributing representations for perceptual schemas across the modules in complicated ways rather than representing individual schema with separate modules.

---

### Decision · Program_Chairs · 2022-01-20

**Decision:**

Reject

**Comment:**

This paper develops a mechanism for learning modular state representations in RL that organize recurring patterns into composable schemas. The approach combines modular RNNs as in RIMs (Goyal et al., 2020) with a dynamic feature attention mechanism. There were a variety of concerns in the initial reviews that were addressed by the authors through a set of clarifications and improved empirical analysis, substantially improving the paper. However, there still remain some issues in clarity of presentation and inconsistent empirical results, especially in the form of clear take-aways from the empirical analysis and broader insights from the paper, as detailed in the individual reviews. The authors are encouraged to take these aspects into consideration in revising their manuscript.